# A Genetically Encoded Dark-to-Bright Biosensor for Visualisation of Granzyme-Mediated Cytotoxicity

**DOI:** 10.3390/ijms241713589

**Published:** 2023-09-02

**Authors:** Christopher Bednar, Sabrina Kübel, Arne Cordsmeier, Brigitte Scholz, Hanna Menschikowski, Armin Ensser

**Affiliations:** Institute of Clinical and Molecular Virology, University Hospital Erlangen, Friedrich-Alexander-Universität Erlangen-Nürnberg, 91054 Erlangen, Germany; christopher.bednar@uk-erlangen.de (C.B.); sabrina.kuebel@uk-erlangen.de (S.K.); arne.cordsmeier@uk-erlangen.de (A.C.); brigitte.scholz@uk-erlangen.de (B.S.); hanna.menschikowski@uk-erlangen.de (H.M.)

**Keywords:** granzyme B, fluorogenic biosensor, T cell cytotoxicity, CAR-T cells

## Abstract

Granzyme B (GZMB) is a key enzyme released by cytotoxic T lymphocytes (CTL) and natural killer (NK) cells to induce apoptosis in target cells. We designed a novel fluorogenic biosensor which is able to assess GZMB activity in a specific and sensitive manner. This cleavage-responsive sensor for T cell activity level (CRSTAL) is based on a fluorescent protein that is only activated upon cleavage by GZMB or caspase-8. CRSTAL was tested in stable cell lines and demonstrated a strong and long-lasting fluorescence signal upon induction with GZMB. It can detect GZMB activity not only by overexpression of GZMB in target cells but also following transfer of GZMB and perforin from effector cells during cytotoxicity. This feature has significant implications for cancer immunotherapy, particularly in monitoring the efficacy of chimeric antigen receptor (CAR)-T cells. CAR-T cells are a promising therapy option for various cancer types, but monitoring their activity in vivo is challenging. The development of biosensors like CRSTAL provides a valuable tool for monitoring of CAR-T cell activity. In summary, CRSTAL is a highly sensitive biosensor that can detect GZMB activity in target cells, providing a means for evaluating the cytotoxic activity of immune cells and monitoring T cell activity in real time.

## 1. Introduction

T cells play an essential role in the immune system. They target infected cells through specific recognition of pathogen-specific peptides presented on major histocompatibility complexes (MHC) by antigen-presenting cells (APC). Exocytosis of cytotoxic granules into the immunological synapse is induced upon binding of the T cell receptor to the corresponding MHC complex carrying a pathogen-specific peptide. Cytotoxic proteins, such as perforin and granzymes, then penetrate the cell membrane of the APC, resulting in activation of multiple caspases and finally apoptosis of the target cell. The identification of T cells encoding target-specific T cell receptors is essential in the research and development of novel immunotherapies against infectious diseases as well as various cancer types. Additionally, a major novel concept in the fight against cancer and infectious diseases has emerged over the past decade: T cells equipped with chimeric antigen receptors (CARs), which are able to recognise target cells independently of MHC antigen presentation. CARs are already the underlying principle in six FDA-approved drugs against B cell acute lymphoblastic leukaemia and other haematological malignancies [1,2]. CD19-CARs further have great potential for the treatment of autoimmune diseases [3,4,5,6]. Moreover, this technology can be applied against a wider spectrum of disease, with research being conducted on CARs targeting solid tumours as well as viral infections, such as HIV, EBV or CMV [7,8,9]. Therefore, the assessment of T cell reactivity under certain conditions and at distinct functional stages has become increasingly important. In this study, we developed a genetically encoded, fluorogenic biosensor for measuring T cell effector function. Previous approaches have relied on flow cytometric analysis of T cell activation marker expression, such as CD69, CD107a, GZMB or perforin; the measurement of secreted cytokines connected to T cell activation, such as TNF-α or IFN-γ via ELISA or ELISPOT assays; or the assessment of overall cytotoxicity via ^51^Cr release, europium release, LDH release or luciferase release assays [10]. A more specific and convenient technique, which is also applicable for high-throughput screening, is the deployment of a fluorogenic dark-to-bright biosensor based on an autofluorescent protein (FP). By taking advantage of proteolytic specificities of certain endogenous proteases and the ability of FPs to change their conformation from a non-fluorescent to a fluorescent state, several biosensors for protease activity have been developed [11,12]. Our own construct shares some similarities with the design of Gong et al. [13], whose reporter indicates the activity of the endogenous enzyme caspase-3. Caspase-3 cleavage activity results in apoptosis of the cell and is also activated downstream during T cell mediated cytotoxicity; however, additional extrinsic apoptotic pathways as well as intrinsic pathways can trigger the activation of caspase-3. GZMB represents a specific protease involved in T cell cytotoxicity and is only present in target cells after exocytosis by activated effector cells. Thus, T-cell-mediated cytotoxicity precisely correlates with GZMB activity in the aforementioned target cells.

## 2. Results

### 2.1. Design of the GZMB-Responsive Sensor

To create a fluorogenic GZMB-responsive reporter, we took advantage of self-splicing split inteins, which bears a similarity to the elegant design of a fluorogenic caspase-3 reporter by Gong et al. [13]. The C-terminal intein of *Nostoc punctiforme* DNA polymerase III (*Npu* DnaEc) was coupled to the C-terminal half of mNeonGreen2 (K159-D229) and separated by a recognition sequence for GZMB (IEPD|SG; [14]) from the N-terminal half of mNeonGreen2 (E7-D158) and the N-terminal intein (*Npu* DnaEn). The C-terminus of the construct contained a proline, glutamate, serine and threonine-rich (PEST) sequence derived from murine ornithine carboxylase (S244-V288) to reduce background fluorescence of unspliced products. In theory, after translation of the protein, the inteins initiate post-translational splicing to create a circular protein while excising the PEST sequence. Thereby, the two mNeonGreen2 halves are fixed into a non-fluorescent state. Upon cleavage of the GZMB recognition site, the intramolecular tension is relaxed and mNeonGreen2 can fold into its native conformation, resulting in the maturation of the chromophore inside of the barrel structure. After chromophore maturation, the correctly folded reporter is able to fluoresce upon excitation with light of around 500 nm wavelength. Since the reporter is activated by GZMB, the major T cell effector protease, we named the construct cleavage-responsive sensor for T cell activity level (CRSTAL, Figure 1). 

Various in vitro assays using ectopically expressed GZMB, as well as killing assays in more native settings, were conducted.

### 2.2. CRSTAL Responds to Ectopic Expression of Active GZMB

In order to investigate the functionality of CRSTAL, the reporter construct was cloned into a lentiviral expression vector and the stable cell line CRSTAL-293T was generated using lentiviral transduction. Contrary to effector caspases that can be activated by addition of inducing agents, such as staurosporine to induce caspase-3 [15], GZMB cannot be induced in CRSTAL reporter cells by adding apoptosis-inducing agents. In cytotoxic lymphocytes, GZMB is expressed as an inactive precursor, which is only activated upon proteolytic cleavage within cytolytic vesicles. The amino acid residues following the signal peptide (glycine 19 and glutamic acid 20; GE) are cleaved off by dipeptidyl-peptidase, resulting in the proteolytic active form of GZMB. Therefore, transfection of a DNA construct encoding full length GZMB would not activate CRSTAL, since the transfected inactive GZMB precursor would not be able to cleave the reporter. Smyth et al. deleted the coding sequence for the inactivation dipeptide GE from an expression construct of GZMB, transfected the resulting plasmid in cells and analysed the respective proteolytic properties [16]. We recreated the precursor (GZMB) and active GZMB (GZMBΔGE) by molecular cloning and used them for qualitative tests of our reporter function (Figure 2a). CRSTAL-293T cells were transfected with pGZMB, pGZMBΔGE or a control vector and analysed 24 h post transfection using the ImageXpress^®^ Pico instrument (Figure 2b). Transfection of CRSTAL-293T with pGZMBΔGE resulted in a strongly increased green fluorescence signal compared to the mock-transfected cells. As expected, cells transfected with the inactive precursor did not show any significant increase in green fluorescence, indicating that GZMBΔGE is able to activate CRSTAL far more strongly than the inactive precursor GZMB.

### 2.3. CRSTAL Is Activated upon Induction of GZMBΔGE Expression

Having proven that CRSTAL is activated by GZMBΔGE, the next step was to eliminate the variable of transfection efficiency by using stable cell lines. Since GZMBΔGE induces apoptosis, it cannot be expressed constitutively. Therefore, inducible cell lines were established that express CRSTAL constitutively and express GZMBΔGE in a doxycycline-dependent manner (CRSTAL-293T-iGZMBΔGE). Cells were seeded on day 1 and treated with 100 ng/mL doxycycline the next day to induce GZMBΔGE expression. On day 3, cells were analysed by fluorescence microscopy using the ImageXpress^®^ Pico (Figure 3a), or lysed and analysed via Western blot (Figure 3b). Doxycycline concentration and incubation period were titrated in advance.

The resulting fluorescence images clearly show a strong increase in green fluorescence in the induced reporter cells expressing GZMBΔGE in comparison to un-induced mock-treated cells (Figure 3a). Western blot analysis extended these results on the molecular level (Figure 3b). Reporter cells treated with doxycycline show the expected 35 kDa band of GZMBΔGE, confirming that the induction of GZMBΔGE expression via doxycycline works flawlessly without leaky expression in untreated cells. Moreover, a clear reduction in the migration distance of CRSTAL in the doxycycline-treated sample is evident: Cleavage of CRSTAL by induced GZMBΔGE linearises the initially circular reporter, causing it to migrate more slowly through the polyacrylamide gel (Figure 3b). 

Taken together, these findings demonstrate the functionality of CRSTAL as an indicator of GZMBΔGE activity on a phenotypical level via induced fluorescence, as well as on a molecular level via cleavage of the reporter protein. Having shown that the reporter functioned in this synthetic model, the next step was to evaluate the temporal kinetics of CRSTAL. 

### 2.4. CRSTAL Elicits a Long-Lasting Fluorescence Signal

In order to analyse the temporal behaviour of CRSTAL, multiple time course experiments were conducted. To evaluate the activation time of CRSTAL upon doxycycline-dependent GZMB induction, CRSTAL-293T-iGZMBΔGE were seeded and induced with 100 ng/mL every 12 h for the time points 36–72 h, and every 2 h for the time points 0–36 h. Cells were harvested after 72 h and analysed via flow cytometry (Figure 4a). The fluorescence of activated cells starts to increase at about 22 h post induction and reaches its maximum after approximately 48 h, with half of the maximum fluorescence reached after 32.9 h. Different fluorescent proteins are known to exhibit maturation times for the formation of the functional chromophore spanning from few minutes to several hours [17]. Since the chromophore of CRSTAL is only able to mature upon cleavage by GZMB and proper folding of the barrel structure, time course experiments assessing the folding and maturation time via induction of GZMBΔGE were conducted. To separate the time required for functional GZMBΔGE to be expressed and subsequent CRSTAL cleavage from the actual maturation time, Western blot analysis was performed. Therefore, CRSTAL-293T-iGZMBΔGE cells were seeded and induced with 100 ng/mL doxycycline in timepoints of every 2 h. Cells were harvested at 24 h and 50 µg cell lysate per lane was separated by SDS-PAGE. Subsequent Western blot analysis was performed using antibodies targeting mNeonGreen, Flag and GAPDH (Figure 4b). Due to different migration distances of cyclised and cleaved CRSTAL, it is possible to differentiate between inactive and active isoforms. GZMBΔGE is detected in the Western blot as early as 2 h post induction with doxycycline and most of the CRSTAL molecules are cleaved. Despite this observation, the fluorescence signal starts to increase 20 h later. In summary, even though CRSTAL is cleaved in a matter of few hours after induction, the reporter takes around 20 h to fold and mature. Nevertheless, CRSTAL shows an approximately 20-fold increase in fluorescence intensity at its peak (48 h after activation), which is stable for a few more days, only losing less than 50% of its maximum fluorescence intensity over the next day.

### 2.5. GZMB Is Sufficient for CRSTAL Activation

The fact that the recognition sequence of GZMB (IEPD) also resembles one recognition sequence of caspase-8 (IETD) led to the assumption that CRSTAL might also be cleavable by caspase-8. Since the GZMB pathway in the cell also activates caspase-8, it is not directly evident whether CRSTAL is mainly activated by GZMB, caspase-8 or by further downstream caspases. To investigate the actual major effector protease, a GZMB-independent caspase-8 pathway was activated via inducible expression of an active caspase-8. For this purpose, CRSTAL-293T-iCasp8FT cells were generated that express the catalytically active subunits p18 and p10: Pro-caspase-8 consists of two death effector domains (DED1 and DED2), as well as the catalytic subunits p18 and p10. Upon stimulation of cells by death ligands, the death-inducing signalling complex (DISC) is recruited and caspase-8 undergoes autocatalytic cleavage at the aspartic acid residue 374 (D374) to release the p10 subunit with a 10-amino-acid N-terminal linker. In the second phase, the linker is cleaved off at D384, and the p18 catalytic subunit is released from the DISC via cleavage at D216. The active form of caspase-8, consisting of the heterotetramer p18_2_-p10_2_, dissociates from the DISC to activate further executioner caspases [18]. To circumvent the necessity of DISC-mediated autocleavage in 293T cells, we designed a self-cleaving version of caspase-8. The N-terminal sequence encoding DED1 and DED2 was deleted and replaced by a Flag tag, followed directly by the p18 subunit. The 10-amino-acid linker sequence between p18 and p10 was replaced with a furin cleavage site (F), a short linker, and a self-cleaving T2A peptide (T), resulting in the name Casp8FT. In this way, the catalytic subunits are separated upon expression, leaving only minor residues on the ends of the fragments (RRKR at the p18 C-terminus; P at the p10 N-terminus) to allow immediate assembly of the catalytically active caspase-8 tetramer (Figure 5a). CRSTAL-293T-iCasp8FT and CRSTAL-293T-iGZMBΔGE cells were seeded and activated using doxycycline in presence or absence of the pan-caspase inhibitor Z-VAD-fmk. Cells were analysed 48 h post activation via flow cytometry (Figure 5b) or via Western blot using antibodies targeting mNeonGreen, GZMB, cleaved caspase-8, and heat shock protein 70 (HSP70) (Figure 5c).

Using this novel doxycycline-inducible active caspase-8 construct, analyses of flow cytometric data showed that CRSTAL activation can be completely inhibited: administration of 100 µM of the pan-caspase inhibitor Z-VAD-fmk in CRSTAL-293T-iCasp8FT cells reduces the MFI fold change from 3.4 to 1.0. Performing the same experiment in a cell line expressing active GZMB, CRSTAL was still activated, but to a reduced degree; the presence of the inhibitor reduced the MFI fold change from 24.4 to 10.4. These results indicate that caspase-8 is sufficient for activation of CRSTAL, but that GZMB cleaved the reporter even when all caspases were inhibited (Figure 5b). The reduced fluorescence in CRSTAL-293T-GZMBΔGE cells treated with Z-VAD-fmk is explained by the inhibition of GZMB-activated endogenous caspase-8, revealing the share of CRSTAL molecules cleaved by GZMB itself. Western blot results confirm this hypothesis, showing that nearly all CRSTAL molecules are cleaved when active GZMB is present, independent of Z-VAD-fmk (Figure 5c). Upon caspase-8 activation in absence of inhibitor, a shift in favour of cleaved CRSTAL is visible. When caspase inhibitor Z-VAD-fmk is added, this effect is abrogated, and CRSTAL band intensities are indifferentiable from uninduced cells. These observations lead to the conclusion that CRSTAL can be cleaved by GZMB as well as by caspase-8, whereas its activation by the latter caspase can be abrogated by the addition of Z-VAD-fmk, making CRSTAL a suitable indicator for GZMB-activity.

### 2.6. CRSTAL Is a Suitable Indicator for CAR-T-Cell-Mediated Cytotoxicity

With the emerging importance of cellular immunotherapies, novel technologies to efficiently analyse the efficacy of the cell products in research and development are required. CAR-T cells constitute a promising therapeutic option and have already proven their efficiency in haematologic malignancies [1,2]. To test whether CRSTAL functions in native settings with actual effector T cells, we designed a CAR construct targeting CD19, an established target for CAR-T cell killing assays. The CAR, called CD19FBBz, was cloned into a murine retroviral expression vector and transduced into murine T cells via spinfection, resulting in CD19FBBz-CTL. CRSTAL-293T-CD19 target cells were generated via lentiviral transduction, and CD19 expression was validated via flow cytometry. CRSTAL-293T-CD19 cells were seeded, and the next day, CD19FBBz-CTL or CAR-negative CTL were added in effector-to-target ratios of 5:1 (Figure 6a,b) and 3:1 (Figure 6c). Cells were co-cultured for 48 h and analysed via flow cytometry (Figure 6a), via ImageXpress^®^ Pico (Figure 6b), or stained with anti-CD8 antibody to be analysed via confocal laser scanning microscope (Figure 6c).

Flow cytometric analyses after 48 h of co-incubation of CRSTAL-expressing target cells and CAR-T cells showed a significant increase in fluorescence intensity (~2.2-fold increase) in comparison to co-incubation with T cells lacking a CAR or with no effector cells at all (Figure 6a). Visual analysis using the ImageXpress^®^ Pico in a 96-well format (Figure 6b) confirmed this observation. Confocal imaging clearly showed target cells with a bright fluorescence of CRSTAL forming apoptotic blebs after being attacked by CAR-T cell in close proximity (Figure 6c). 

Previous studies on the substrate specificity of human and murine GZMB have shown that human caspase-8 is not cleaved by murine GZMB [19], adding to the observation that GZMB is sufficient to activate CRSTAL independently from caspase-8, since in the setting of this experiment (murine T cells versus human target cells), caspase-8 is not activated by GZMB during the killing process. 

## 3. Discussion

In this work, we designed a highly sensitive and stable dark-to-bright reporter for GZMB activity based on the fluorescent protein mNeonGreen2. We were able to show that although CRSTAL is cleaved by GZMB as well as by caspase-8, GZMB activity is sufficient for activation of the sensor. This makes the reporter a suitable indicator for effector-cell-mediated cytotoxicity, caused by the exclusive presence of active GZMB in target cells after introduction by T or NK cells. The specificity of the CRSTAL signal exclusively induced by GZMB can be achieved by addition of Z-VAD-fmk to inhibit caspase-8 activity. CRSTAL is cleaved efficiently in a matter of minutes to a few hours, whereas the folding and/or maturation of the fluorophore is a process lasting several hours (MFI_50_ = 32.9 h). We also showed that CD19-specific CAR-T cells were able to recognise and lyse CD19-expressing reporter cells, activating CRSTAL in the process. Our reporter exerts a GZMB-specific, long-lasting and stable fluorescence signal over multiple days without losing much of its fluorescence intensity. These criteria make CRSTAL a suitable biosensor for multiple immunological approaches, and facilitates numerous assays related to T or NK cell activity.

Conventional methods for assessing CTL-mediated cytotoxicity, such as ^51^Cr release, europium release, LDH release or luciferase release assays have several drawbacks compared to our biosensor. First of all, ^51^Cr release uses radioactive isotopes which are released from damaged cells. Many laboratories have abolished their isotope work due to the emergence of less harmful alternatives and very expensive waste management. Europium release assays require the loading of target cells with europium, which is, on one hand, fairly expensive, and on the other hand, an additional time-consuming step in the protocol. LDH release does not require pre-incubation of cells with harmful or expensive reagents; however, it is not possible to differentiate the LDH released from target cells from the LDH released by effector cells. Luciferase release assays are based on target cells stably expressing some form of luciferase which is released into the supernatant upon killing by effector cells [20]. This approach can circumvent the unknown origin problem of LDH release assays, but as the other approaches, this assay has more drawbacks when a third party, such as viruses, comes into play. When considering T cell responses against virus-infected target cells, our reporter can solve two issues that all of the aforementioned assays inherit. In such T cell killing assays, not only are the effector cells able to lyse the cells, but concurrent viral cytolysis makes it difficult to differentiate T-cell-mediated killing from virus-mediated killing. On the other hand, several viruses also exert anti-apoptotic effector functions to circumvent T-cell-mediated cytotoxicity and prevent premature death of the host cell. These effector proteins, such as UL36 and UL37x1 from the human cytomegalovirus, mostly target host cell mechanisms downstream of GZMB cleavage of caspase-8 [21]. Therefore, even when viral effector proteins inhibit apoptosis of the target cell, as shown by our group [22,23], our CRSTAL reporter is able to recognise CTL-induced GZMB activity.

The intracellular detection of GZMB activity using the methods described in this study requires the introduction of CRSTAL into target cells via transient transfection or stable transduction, making it a suitable biosensor for in vitro assays. It would be possible to analyse GZMB activity in the animal model, using either genetically modified mice expressing CRSTAL, or using xenograft models in which reporter-expressing target cells have been implanted. However, monitoring GZMB activity in patients is only possible when the cytotoxic potential of patients’ T cell is assessed ex vivo in cell culture. An alternative approach is small fluorescent molecules coupled via GZMB-sensitive linker peptides to a quencher; these sensors are activated upon cleavage by an effector protease, separating the quencher from the fluorophore resulting in a fluorescence signal [24,25,26,27]. These approaches, given tolerable side effects, might enable the monitoring of GZMB activity in patients. The functionality of these probes was already shown in murine models [24]. However, the artificial nature of these fluorogenic probes will require extensive testing for toxicity or unwanted immunologic reactions before administration to the patient.

With the emergence of CAR-T cell therapy, it is becoming more and more important to optimise the design of a CAR for its intended use. Therefore, a convenient readout of T cell cytotoxicity, such as CRSTAL, can facilitate the time-consuming process of screening through numerous constructs. By using ingenious cloning techniques, it is possible to create libraries containing multiple different building blocks for CAR domains, such as different signal peptides, single-chain variable fragments (scFv), hinge domains, transmembrane domains, costimulatory domains and activating domains [28,29]. After assembly and transduction into suitable effector T cells, CRSTAL-expressing reporter cells can be used to screen for the brightest signals in green fluorescence to identify T cells expressing the specific CAR constructs that lead to the strongest CTL-mediated cytotoxicity. In conclusion, various readout options can be used with CRSTAL as a reporter for CAR-T cell activity analyses, making this sensor an attractive option for time-saving high-throughput screening in a 96-well format without additional staining using flow cytometry or high content screening instruments. 

## 4. Materials and Methods

### 4.1. Design of DNA Constructs

The GZMB-inducible biosensor described in this paper is loosely based on the construct of Gong et al. [13]. The construct was designed in silico and synthesised de novo as a human-codon-optimised fragment. The functional rationale behind this construct is based on the ability of FPs to refold from a non-fluorescent state to a fluorescent conformation. Two split halves of a FP are forced into the non-fluorescent conformation by cyclisation of the protein using split-inteins which undergo post-translational splicing. Separated by a specific protease recognition site, the malformed FP is only able to switch into a fluorescent conformation upon cleavage by the respective protease. Here, we use mNeonGreen2 (GenBank: AGG56535) as the functional core of the GZMB sensor. The FP is split into two truncated halves corresponding to the two parts mNeonGreen2_K159-D229_ and mNeonGreen2_E7-D158_. These two halves are separated by the GZMB-cleavable oligopeptide IEPDSG, which is recognised by GZMB and cleaved after the aspartate residue [14]. The N-terminal domain is formed by the intein *Npu* DnaEc (GenBank: WP_012411929.1; M1-N39) of the DNA polymerase III (DnaE) of the bacterium *Nostoc punctiforme*. To undergo post-translational splicing, an intein counterpart is required. Therefore, mNeonGreen2_E7-D158_ is followed by the sequence encoding for *Npu* DnaEn (GenBank: WP_012411174.1; A769-N874). To reduce the background fluorescence of unspliced proteins, the construct ends in a PEST sequence of the murine ornithine carboxylase (GenBank: AAA39846.1; amino acids S244-V288), resulting in an accelerated degradation of unspliced sensor proteins [13]. The finished construct looks as follows: *Npu* DnaEc-mNeonGreen2_K159-D229_-IEPDSG-mNeonGreen2_E7-D158_-*Npu* DnaEn-PEST (Appendix A). The whole construct is flanked by sequences complementary to the insertion site of the lentiviral expression plasmid pLV-EF1α-IRES-Hygro (Addgene: #85134) to facilitate cloning via Gibson assembly [30].

### 4.2. Molecular Cloning

The lentiviral vectors pLV-EF1α-IRES-Hygro (#85134), pLenti-CMVtight-Puro-DEST (#26439) and pLenti-CMV-rtTA3-Blast (#26429) were obtained from Addgene (Watertown, MA, USA; Appendix A). The lentiviral helper plasmids psPAX2 and pMD2.G were kindly provided by Prof. Didier Trono, École Polytechnique Fédérale de Lausanne. The mammalian expression vector pCDNA3.1 was purchased from Invitrogen (Carlsbad, CA, USA) and digested with BamHI and Acc65I. Primers and DNA fragments (gBlocks^®^) were synthesised by Integrated DNA Technologies (Coralville, ID, USA). Using the primers BamHIGZMB-fwd and GZMBFlagAcc65I-rev (Appendix A), the transcript of human GZMB was amplified from human cDNA, adding the coding sequence for a C-terminal Flag tag to the 3′ terminus of the gene. The resulting PCR product was digested with BamHI and Acc65I and ligated into the cleaved pCDNA3.1 vector backbone, resulting in the construct pGZMB. Using the primers GZMBdGE-fwd and GZMBdGE-rev, the codons for the amino acids G19 and E20 were deleted via mutagenesis PCR, resulting in the plasmid pGZMBΔGE. Restriction cloning of pLenti-CMVtight-Puro-DEST and pGZMBΔGE with BamHI and PmeI resulted in the construct pLCTP-GZMBΔGE. To generate the inducible active caspase-8 construct, a DNA fragment was designed consisting of an N-terminal Flag tag followed by the coding sequence of human caspase-8 S217 to D379 and L385 to D479 containing the catalytically active subunits p18 and p10, separated by the furin cleavage site RRKR, a GSG linker and the T2A self-cleaving peptide. It was synthesised de novo as a codon-optimised fragment and assembled via Gibson assembly into the vector pLenti-CMVtight-Puro-DEST digested by BamHI and PmeI to result in the plasmid pLCTP-iCasp8FT. Via PCR mutagenesis using the primers ATG-Kozak-Not-as and Stop-Pme-LV-IRES, pLV-EF1α-IRES-Hygro and pLV-EF1α-IRES-Puro were linearised. The fragment encoding the biosensor CRSTAL, flanked by complementary regions to the opened backbone, was synthesised de novo and assembled into the linearised pLV-EF1α-IRES-Hygro vector by Gibson-Assembly resulting in the plasmid pLV-EIH-CRSTAL. Using the primers pLV-CD19-fwd and pLV-CD19-rev, the transcript of human CD19 was amplified from human cDNA. The resulting PCR product was assembled into the linearised backbone of pLV-EF1α-IRES-Puro resulting in the construct pLV-EIP-CD19. 

The CD19-specific CAR CD19FBBz consists of the CD8 signal peptide and the single-chain variable fragment (scFv) of the CD19 antibody clone FMC63, separated by a (G_4_S)_3_ linker. The sequence for a Flag tag is located between the scFv and the transmembrane domain of CD8. The co-stimulatory domain of 4-1BB and the activating domain CD3ζ make up the C-terminal end of the CAR. The whole amino acid sequence can be found in the Appendix A. The coding sequence for CD19FBBz was synthesised de novo as a DNA fragment containing overlaps and assembled into the multiple cloning site of the retroviral expression vector pMSCV-IRES-Puro resulting in the plasmid pMSCV-CD19FBBz. The amino acid sequences of constructs generated in this study are shown in Appendix A.

### 4.3. Cell Culture

All cells were cultivated at 37 °C, 5% CO_2_, and 80% humidity. Human embryonic kidney (HEK) 293T cells (ATCC^®^ CRL-3216™, ATCC, Manassas, VA, USA) were cultivated in Dulbecco’s Modified Eagle Medium (DMEM; 11500516, Thermo Fisher Scientific, Waltham, MA, USA) supplemented with 10% heat-inactivated foetal bovine serum (FBS; 10270106, Thermo Fisher Scientific, Waltham, MA, USA), 50 µg/mL gentamicin (1405-41-0, Serva Electrophoresis, Heidelberg, Germany), 2 mM GlutaMAX™ (35050061, Thermo Fisher Scientific) and 25 mM HEPES (15630080, Thermo Fisher Scientific). Primary murine T cells were cultivated in Roswell Park Memorial Institute (RPMI) 1640 (11875093, Thermo Fisher Scientific) supplemented with 10% heat-inactivated FBS, 50 µg/mL gentamicin, 2 mM GlutaMAX™, 25 mM HEPES, 1 mM sodium pyruvate (11360070, Thermo Fisher Scientific), 50 µM β-mercaptoethanol (A1108, AppliChem, Darmstadt, Germany), 50 IU/mL rhIL-2 (11147528001, Merck, Darmstadt, Germany). HEK293T cells stably expressing CRSTAL (CRSTAL-293T) were cultivated in complete DMEM additionally supplemented with 200 µg/mL hygromycin B (H0654, Merck). HEK293T cells stably expressing CRSTAL and CD19 (CRSTAL-293T-CD19) were cultivated in complete DMEM additionally supplemented with 200 µg/mL hygromycin B and 2 µg/mL puromycin (SC-1080713, Santa Cruz Biotechnology, Dallas, TX, USA). Tetracycline-inducible cells (CRSTAL-293T-iGZMBΔGE, CRSTAL-293T-iCasp8FT) were cultivated in tetracycline-free complete DMEM additionally supplemented with 2 µg/mL puromycin, 5 µg/mL blasticidin (ANT-BL-1, InvivoGen, San Diego, CA, USA) and 200 µg/mL hygromycin B. The retroviral packaging cells Platinum-E (RV-101, Cell Biolabs/BioCat Heidelberg, Germany; PlatE [31]) were cultured in complete DMEM additionally supplemented with 1 µg/mL puromycin and 10 µg/mL blasticidin.

### 4.4. Antibodies

The following antibodies were used for Western blot, confocal microscopy and flow cytometric analyses: Anti-Flag^®^BioM2 (mouse, #F9291, Merck), Anti-mNeonGreen (rabbit, #53061, Cell Signalling Technologies, Danvers, MA, USA), biotin-conjugated anti-GAPDH (goat, #A00915, Genscript, Piscataway, NJ, USA), anti-human cleaved caspase-8 (mouse, #sc-5263, Santa Cruz Biotechnologies), anti-human granzyme B (mouse, #sc-8022, Santa Cruz Biotechnologies), anti-human HSP 70 (mouse, #sc-24, Santa Cruz Biotechnologies), AlexaFluor^®^647-conjugated anti-rabbit IgG (goat, #A32733, Thermo Fisher Scientific), AlexaFluor^®^555-conjugated anti-mouse IgG (goat, #A21422, Thermo Fisher Scientific), APC-conjugated anti-mouse CD8a (rat, #100712, BioLegend, San Diego, CA, USA) and DyLight^®^549-conjugated streptavidin (#A21837, Thermo Fisher Scientific).

### 4.5. Transfection

HEK293T or PlatE cells were transfected using polyethylenimine (23966-100, Poly-sciences Inc, Warrington, PA, USA; PEI; 1 mg/mL). Cells were seeded the day before transfection. On the next day, the appropriate amount of plasmid DNA was diluted in OptiMEM (31985062, Thermo Fisher Scientific). PEI was diluted in the same amount of OptiMEM in a relation of µg DNA: µL PEI = 1:3. DNA dilution and PEI dilution were mixed, incubated for 15 min at room temperature and added dropwise to the cells. Medium was exchanged six hours post transfection.

### 4.6. Production of Lentiviral Particles

On the day before transfection, HEK293T cells were seeded in a T25 cell culture flask. The cells were transfected with a total amount of 5 µg DNA comprised of lentiviral expression plasmid as well as the lentiviral packing vectors psPAX2 and pMD2.G in a stochiometric relation of 3:1:1, as described in Section 4.5. Three days after transfection, the lentivirus-containing supernatant was separated from cell debris by centrifugation (1000× *g*, 5 min). Lentivirus-containing supernatant was used immediately or stored at −80 °C until further usage.

### 4.7. Lentiviral Transduction of Adherent Cells

HEK293T cells were seeded in a 6-well plate one day before transduction. On the next day, 1 mL of lentivirus-containing supernatant was added to each well, and polybrene (H9268, Merck) was added to a final concentration of 10 µg/mL. The cells were incubated for 48–72 h under cell culture conditions resulting in the transduced cell lines CRSTAL-293T, CRSTAL-293T-iGZMBΔGE, CRSTAL-293T-iCasp8FT and CRSTAL-293T-CD19. The cells were selected in their respective culture medium supplemented with the according concentration of antibiotics Section 4.3 and were ready for usage five days after selection.

### 4.8. Generation of CD19-CAR-T Cells

CAR-T cells were generated as follows: PlatE cells were seeded in a 6-well plate transfected with 2 µg pMSCV-CD19FBBz + 1 µg pCL-Eco on the next day, as described in Section 4.5. Two and three days after transfection, the supernatant was harvested, combined and separated from cell debris by centrifugation. Aliquots were used immediately or stored at −80 °C until further usage. A 12-well plate was coated with goat anti-hamster polyclonal antibody (5 µg/mL in PBS, 31115, Thermo Fisher Scientific) at 4 °C overnight. The next day, murine splenocytes were isolated from mouse spleens using ammonium chloride erythrocyte lysis. T cells were activated in complete RPMI supplemented with 50 U/mL recombinant human (rh) IL-2 and anti-CD3 (1 µg/mL, 100302, BioLegend) and anti-CD28 (1 µg/mL, 102102, BioLegend) antibodies at a cell density of 3 × 10^6^ T cells per mL in the coated 12-well plate and incubated under cell culture conditions. The next day, T cell media was aspirated and stored at 4 °C and PlatE supernatant containing CD19-CAR-encoding retrovirus was loaded onto the T cells. Polybrene was added to a final concentration of 10 µg/mL. The plate was centrifuged at 800× *g* for 1.5 h at 32 °C with moderate acceleration and deceleration. The cells were incubated for 2–4 h under cell culture conditions. The medium was exchanged for a 1:1 mixture of the previously stored supernatant with fresh complete RPMI supplement containing 50 IU/mL rhIL-2, but no antibodies, and incubated overnight. The medium was exchanged for fresh complete RPMI supplemented with 50 IU/mL rhIL-2 and incubated overnight. Two days after transduction, CAR expression was evaluated by flow cytometry using an anti-Flag antibody. CAR^+^ cells were selected and expanded in complete RPMI supplemented with 7.5 µg/mL puromycin 50 IU/mL rhIL-2, 10 ng/mL rhIL-7 (#200-07, PeproTech, Hamburg, Germany) and 10 ng/mL recombinant murine (rm) IL-15 (#210-15, PeproTech) at a density of 1–3 × 10^6^ CAR^+^ cells per mL for 3–5 days. CAR expression was evaluated by flow cytometry.

### 4.9. Time Course Measurements

Flow cytometry: CRSTAL-293T-iGZMBΔGE cells were seeded in 24-well plates and induced with 100 ng/mL doxycycline every 12 h for the first 36 h and every 2 h for the following 36 h. Cells were harvested, washed once in FACS buffer (PBS + 5% FCS), and analysed in the Attune^®^ NxT flow cytometer (Life Technologies, Carlsbad, CA, USA). Western blot: CRSTAL-293T-iGZMBΔGE cells were seeded in 24-well plates and induced with 100 ng/mL doxycycline every 2 h for 24 h. Cells were harvested, washed once, and lysed as described in Section 4.10.

### 4.10. Western Blot

Cells were harvested and washed with PBS. Cell lysis was performed for 30 min on ice using 100 µL RIPA buffer (10 mM Tris-HCl, 150 mM NaCl, 1% IGEPAL, 0.5% sodium deoxycholate, pH 8.0). The lysate was centrifuged at 20,000× *g* for 30 min at 4 °C and the protein concentration was determined using the Pierce™ BCA Protein Assay Kit (23227, Thermo Fisher Scientific) according to the manufacturer’s instructions. Lysates containing 50 µg of total protein were mixed with 4x Roti^®^-Load (K929.1, Carl Roth, Karlsruhe, Germany), denatured at 95 °C for 5 min and loaded onto a NuPAGE™ 4–12% Bis-Tris Gel (NP0321PK2, Thermo Fisher Scientific), and gel electrophoresis was performed for 90 min at a constant voltage of 130 V. The proteins were blotted onto a methanol-activated Immobilon^®^-FL PVDF membrane (Merck) for 90 min at constant amperage of 60 mA. The membrane was blocked using 5% skim milk powder in PBS + 0.1% Tween™ 20 (A4972, Applichem, Darmstadt; PBST) for 30 min. Proteins of interest were detected using antibodies targeting mNeonGreen, Flag, GZMB, cleaved caspase-8, cleaved caspase-3, GAPDH and HSP70. Primary antibodies were used at a 1:1000 dilution in NET gelatine (1.5 M NaCl, 50 mM EDTA, 0.5 M Tris-HCl pH 7.5, 0.5% Triton X-100, 25 g/L gelatine) and incubated on the blot overnight at 4 °C. After five washes with PBST, secondary antibodies were applied at 1:2000 dilution in NET gelatine for 1 h at RT. Blots were washed five times and visualised using the INTAS advanced fluorescence imager (Intas Science Imaging Instruments GmbH, Göttingen, Germany). 

### 4.11. Flow Cytometry

Cells were harvested and washed once in FACS buffer. Primary antibodies were diluted in FACS buffer. After 1 h of incubation with primary antibodies, the cells were washed twice and stained with the respective secondary antibodies diluted in FACS buffer for 1 h. After two final wash steps, the cells were analysed using the Attune^®^ NxT Acoustic Focusing Cytometer.

### 4.12. Caspase-8 Induction Assay

CRSTAL-293T-iGZMBΔGE and CRSTAL-293T-iCasp8FT cells were seeded in a 24-well plate and treated with doxycycline (1–10,000 ng/mL) in presence or absence of the pan-caspase inhibitor Z-VAD-fmk (100 µM; HY-16658B, MedchemExpress, Monmouth Junction, NJ, USA) the next day. Fluorescence was measured 48 h post treatment via flow cytometry, or cells were lysed and analysed via Western blot using antibodies targeting mNeonGreen, GZMB, cleaved caspase-8 and HSP 70.

### 4.13. CAR-Killing Assay

Target and effector cells were generated as described above (Section 4.8). Flow cytometry: 2 × 10^4^ CRSTAL-293T-CD19 cells were seeded in a flat-bottom 96-well plate. Twenty-four hours later, 1 × 10^5^ CD19FBBz-CAR-T cells were added onto the target cells and incubated for 48 h. The cells were washed and stained with the Fixable Viability Dye eFluor780 (65-0865-14, Thermo Fisher Scientific) as well as anti-CD19-Alexa Fluor^®^647 in FACS buffer for 30 min. The fluorescence intensity of CRSTAL was analysed. Confocal imaging: 1 × 10^4^ CRSTAL-293T-CD19 cells were seeded in a flat-bottom 96-well µ-plate (89626, ibidi, Gräfelfing, Germany). On the next day, CD19FBBz-CAR-T cells were stained with anti-CD8-APC antibody for 30 min at 37 °C. 3 × 10^4^ untransduced T cells or CD19FBBz-CAR-T cells were added onto the target cells, respectively. After one hour of co-incubation, 300 µL overlay media (1% (*w*/*v*) agarose in PBS; boiled and cooled to 40 °C) was carefully layered onto the cell culture. Two days later, cells were visualised with the confocal laser scanning microscope Leica TCS SP5 equipped with a 63 × 1.4 HCX PL APO CS oil immersion objective (Leica Microsystems, Wetzlar, Germany) and analysed with the Leica Application Suite Advanced Fluorescence (Leica Microsystems).

## Figures and Tables

**Figure 1 ijms-24-13589-f001:**
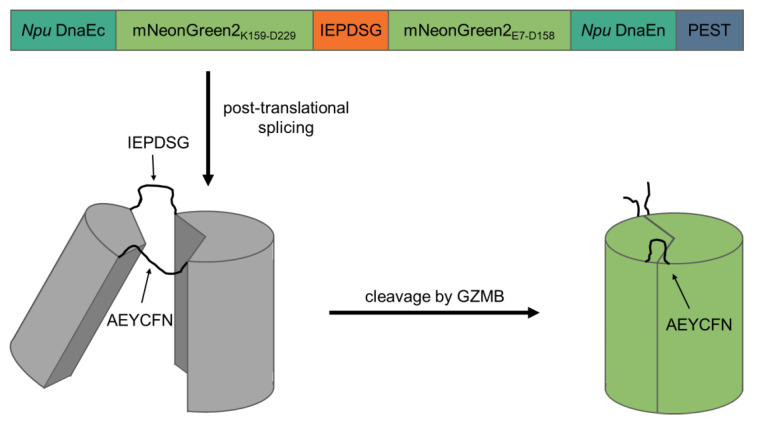
Design and mechanism of CRSTAL. Two parts of the fluorescent protein mNeonGreen2 are separated by the GZMB recognition sequence IEPDSG and flanked by the two inteins of *Npu* DnaE and a C-terminal PEST sequence. Post-translational protein splicing results in cyclisation and subsequent disruption of the barrel structure of mNeonGreen2. The inteins are spliced out, leaving a circular protein consisting of two halves of mNeonGreen2 connected by the splice residues AEYCFN on one side, as well as the GZMB cleavage site IEPDSG on the other side. Unspliced constructs are destabilised by the PEST sequence. Upon specific cleavage of the recognition sequence by GZMB, the reporter is re-linearised and undergoes conformation changes to an excitable state.

**Figure 2 ijms-24-13589-f002:**
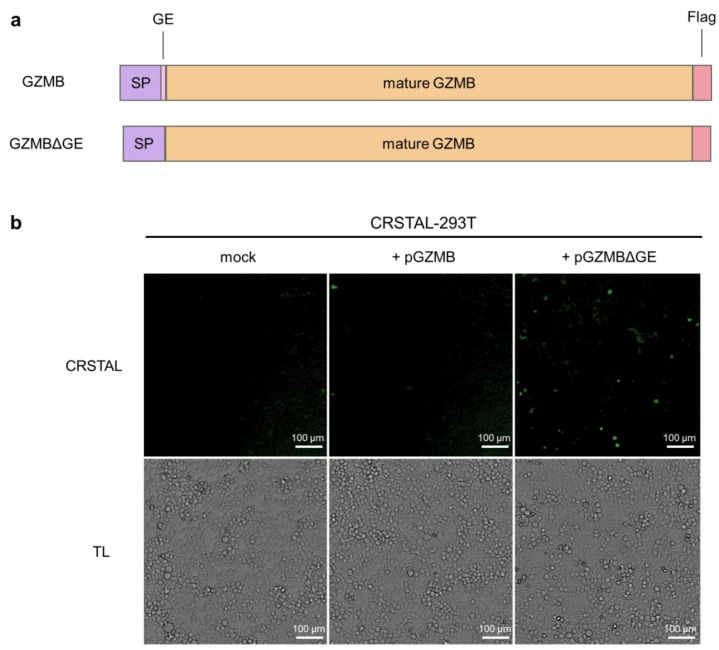
Activation of CRSTAL by ectopic GZMB expression. (**a**) Structure of the constructs of full-length GZMB containing the inactivation peptide GE as well as the mutated version lacking GE (GZMBΔGE); SP signal peptide. (**b**) CRSTAL-293T cells were transfected with pGZMB, pGZMBΔGE or a mock plasmid. CRSTAL fluorescence images and transmitted light (TL) images were acquired and analysed using the ImageXpress^®^ Pico instrument and Cell Reporter express software 2.9.

**Figure 3 ijms-24-13589-f003:**
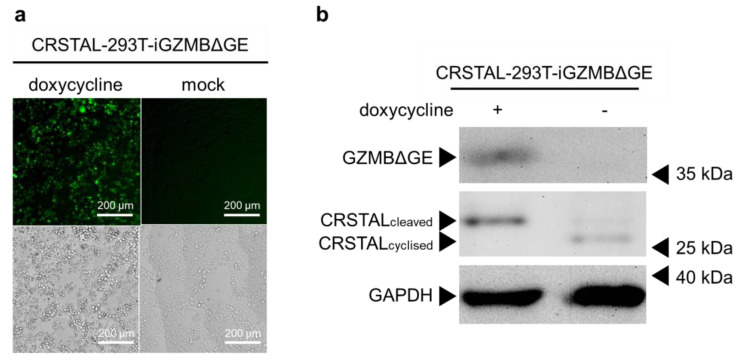
General functionality test of CRSTAL. CRSTAL-293T-iGZMBΔGE were seeded and treated with doxycycline for 48 h. The cells were either (**a**) analysed via fluorescence microscopy or (**b**) lysed and analysed via Western blot.

**Figure 4 ijms-24-13589-f004:**
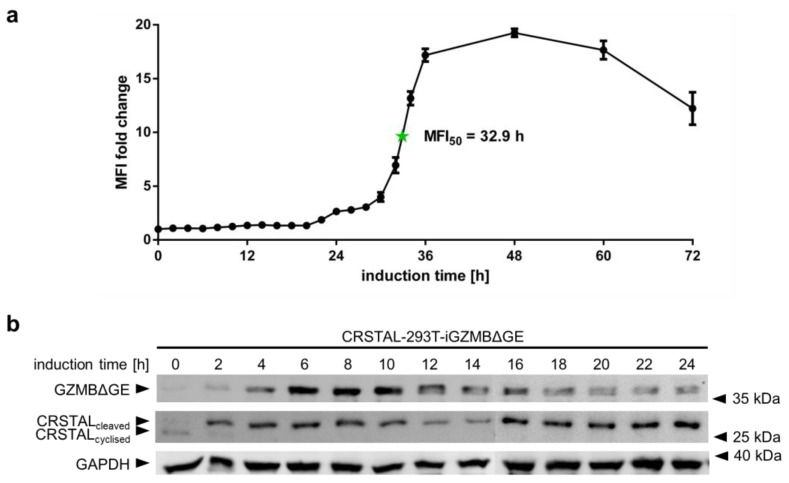
Time course of CRSTAL fluorescence. (**a**) CRSTAL-293T-iGZMBΔGE cells were seeded and induced with 100 ng/mL doxycycline every 12 h for the last time points, then every 2 h for the time points 0–36 h. Fluorescence was measured by flow cytometry. (**b**) CRSTAL-293T-iGZMBΔGE cells were induced with 100 ng/mL doxycycline in steps of 2 h over 24 h. Cells were lysed at 24 h and analysed via Western blot.

**Figure 5 ijms-24-13589-f005:**
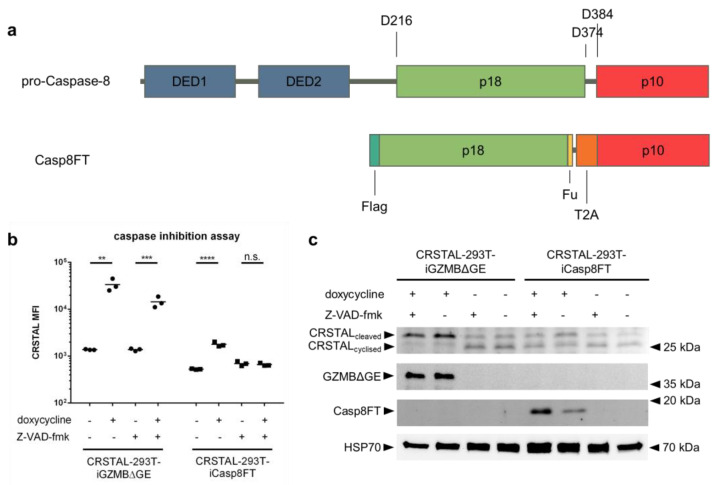
Caspase-8-independent activation of CRSTAL. (**a**) Design of the inducible, self-cleaving variant of caspase-8 (iCasp8FT) based on the sequence of pro-caspase-8. (**b**,**c**) Expression of active caspase-8 or GZMB was induced in CRSTAL-293T-iCasp8FT or CRSTAL-293T-iGZMBΔGE via treatment with 1 µg/mL doxycycline in presence or absence of 100 µM Z-VAD-fmk. Cells were analysed 48 h post induction (**b**) via flow cytometry or (**c**) via Western blot. Statistics: Student’s *t*-test; **: *p* < 0.01; ***: *p* < 0.005; ****: *p* < 0.001; n.s.: not significant. Detailed flow cytometry information is depicted in Appendix A.

**Figure 6 ijms-24-13589-f006:**
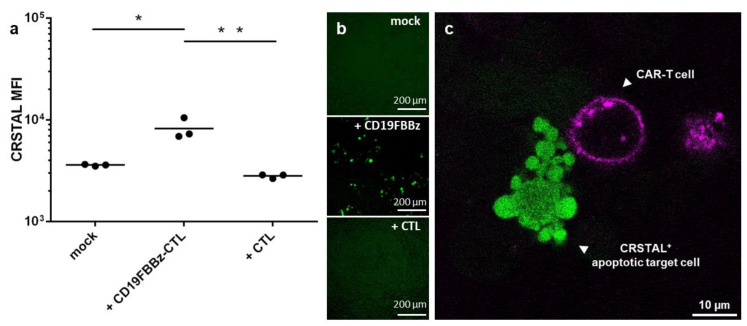
CAR-T cell killing assay on CRSTAL-expressing target cells. CRSTAL-293T-CD19 cells were seeded and co-incubated with CD19FBBz-CTL or CAR-negative CTL at effector-to-target ratios of (**a**,**b**) 5:1 or © 3:1 for 48 h. Cells were stained and analysed (**a**) via flow cytometry, (**b**) analysed via ImageXpress^®^ Pico, or (**c**) via confocal laser scanning microscopy. Statistics: Student’s *t*-test; *: *p* <0.05; **: *p* < 0.01. Detailed flow cytometry information is depicted in Appendix A.

## Data Availability

Not applicable.

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
