# Peer review of "A Genetically Encoded Dark-to-Bright Biosensor for Visualisation of Granzyme-Mediated Cytotoxicity"

_ijms, 2023, doi:10.3390/ijms241713589_

Round 1

Reviewer 1 Report

This manuscript is dedicated to creation of biosensor detecting active forms of Granzyme B. This study and manuscript is well-done and represent the interest for the readers of the journal. 

Comments/suggestions 

- Point 2.2 Quantitative data regarding the difference between fluorescence of mock, pGZMB and pGZMBΔGE trasduced cells is needed

- The limitation of the developed sensor regarding specificity should be desribed in more details

- Please provide flow cytometry data in suplementary materials 

Author Response

We thank the Reviewer 1 who found that our study and manuscript are well-done and represent the interest for the readers of the journal.

Regarding the specific points/comments/suggestions

- Point 2.2 Quantitative data regarding the difference between fluorescence of mock, pGZMB and pGZMBΔGE transduced cells is needed

In 2.2 we qualitatively assed the function of the reporter (now stated in the text). The quantitative analysis is shown in the subsequent paragraphs.

- The limitation of the developed sensor regarding specificity should be described in more details

We have added a paragraph (also in response to reviewer 2) describing limitations and alternatives to our sensor,

lines #285-286:

Specificity of CRSTAL signal exclusively induced by GZMB can be achieved by addition of Z-VAD-fmk to inhibit caspase-8 activity.

lines #319-332:

The intracellular detection of GZMB activity using methods described in this study requires the introduction of CRSTAL into target cells via transient transfection or stable transduction, making it a suitable biosensor for in vitro assays. It would be possible to analyse GZMB activity in the animal model, using either genetically modified mice which express CRSTAL or using xenograft models in which reporter-expressing target cells are implanted. However, monitoring GZMB activity in patients is only possible when the cytotoxic potential of patients T cell is assessed ex vivo in cell culture. An alternative approach are small fluorescent molecules, coupled via GZMB-sensitive linker peptides to a quencher; these sensors are activated upon cleavage by an effector protease, separating the quencher from the fluorophore resulting in a fluorescence signal [24–27]. These approaches, given tolerable side effects, might enable the monitoring of GZMB activity in patients. The functionality of these probes was already shown in murine models [24]. However, the artificial nature of these fluorogenic probes will require extensive testing for toxicity or unwanted immunologic reactions before administration to the patient.

- Please provide flow cytometry data in supplementary materials

We now provide the flow cytometry data and gating strategy in the supplementary file

Reviewer 2 Report

Bednar et al. described a new turn-on fluorescence probe for Granzyme B activity. The sensor protein was consistent of two parts separated by a Granzyme specific peptide sequence, which upon cleavage by granzyme B can fold into a fluorescent protein. The authors demonstrated this Granzyme B sensor in granzyme B transfected cells and in T-cell mediated cytotoxicity in CAR-T therapy. I recommend this manuscript for publication after the following minor revisions:

1. The author mentioned several existing methods for detecting granzyme B activity but left out fluorescent probes. For example ACS Nano 2022, 16, 11, 19328–19334, ACS Cent. Sci. 2022, 8, 5, 590–602. The authors should also discuss the disadvantages of using the method described in this paper for example this method requires generating cells lines express the reporter protein. While using a fluorescent probe is more convenient and does not require using engineered cell lines.

2. For figure 2b, can the authors also add brightfield images and also explain the signal seen when added granzyme B precursor in the middle panel.

3. In figure 4b, why does the active granzyme b increases and then decreases as shown in Wester blot? I thought as dox was added every two hour the induction of active granzyme B should be relatively consistent.

4. Please add scale bars to cell images.

Author Response

The thank the reviewer #2 who recommended our manuscript for publication after the following minor revisions:

  1. The author mentioned several existing methods for detecting granzyme B activity but left out fluorescent probes. For example ACS Nano 2022, 16, 11, 19328–19334, ACS Cent. Sci. 2022, 8, 5, 590–602. The authors should also discuss the disadvantages of using the method described in this paper for example this method requires generating cells lines express the reporter protein. While using a fluorescent probe is more convenient and does not require using engineered cell lines.

We have added a paragraph (also in response to reviewer 1) describing limitations and alternatives to our sensor, lines #319-332:

The intracellular detection of GZMB activity using methods described in this study requires the introduction of CRSTAL into target cells via transient transfection or stable transduction, making it a suitable biosensor for in vitro assays. It would be possible to analyse GZMB activity in the animal model, using either genetically modified mice which express CRSTAL or using xenograft models in which reporter-expressing target cells are implanted. However, monitoring GZMB activity in patients is only possible when the cytotoxic potential of patients T cell is assessed ex vivo in cell culture. An alternative approach are small fluorescent molecules, coupled via GZMB-sensitive linker peptides to a quencher; these sensors are activated upon cleavage by an effector protease, separating the quencher from the fluorophore resulting in a fluorescence signal [24–27]. These approaches, given tolerable side effects, might enable the monitoring of GZMB activity in patients. The functionality of these probes was already shown in murine models [24]. However, the artificial nature of these fluorogenic probes will require extensive testing for toxicity or unwanted immunologic reactions before administration to the patient.

  1. For figure 2b, can the authors also add brightfield images and also explain the signal seen when added granzyme B precursor in the middle panel.

We have retaken pictures with additional transmitted light pictures in Fig 2b.
The CRSTAL signal seen in few cells in the controls (in mock and inactive granzyme B) is likely the result of occasional spontaneous cell death with concomitant activation of e.g. Caspase 8 or other proteases. 

  1. In figure 4b, why does the active granzyme b increases and then decreases as shown in Western blot? I thought as dox was added every two hour the induction of active granzyme B should be relatively consistent.

We are sorry for our phrasing that may have caused a misunderstanding. Cells were kept under induction for specific times, performed in steps of 2h; fresh dox was not added every two hours.
The apparent peak of active GZMB may be explained by the fact that high active GZMB is detrimental to the cell viability. Cells expressing efficiently after induction make more GZMB but are lost, cells surviving to later timepoints may only do so by making less GZMB per cell, or by cells that are less able to be induced - thereby being able to survive; in total this will result in a lower GZMB expression per µg lysate at later timepoints.

  1. Please add scale bars to cell images.

We have added scale bars to cell images